# Ergonutrition Supplementation and Recovery in Water Polo: A Systematic Review

**DOI:** 10.3390/nu17081319

**Published:** 2025-04-10

**Authors:** Álvaro Miguel-Ortega, Josu Barrenetxea-Garcia, María-Azucena Rodríguez-Rodrigo, Enrique García-Ordóñez, Juan Mielgo-Ayuso, Julio Calleja-González

**Affiliations:** 1Faculty of Education, Alfonso X ‘El Sabio’ University (UAX), 28691 Madrid, Spain; 2Regional Ministry of Castilla y León Board of Education, HS Conde Diego Porcelos, 09006 Burgos, Spain; mrodriguezrrod@educa.jcyl.es; 3Water Polo Club Leioa Waterpolo, 48940 Leioa, Spain; jbarrenetxeawp@gmail.com; 4Faculty of Education and Sports Sciences, Vigo University, 36005 Pontevedra, Spain; kikewp@uvigo.gal; 5Faculty of Health Sciences, University of Burgos (UBU), 09001 Burgos, Spain; jfmielgo@ubu.es; 6Physical Education and Sports Department, Faculty of Education and Sport, University of the Basque Country (UPV/EHU), 01007 Vitoria, Spain; 7Faculty of Kinesiology, University of Zagreb, 10110 Zagreb, Croatia

**Keywords:** water polo, nutrition, recovery, performance, ergogenic aids

## Abstract

Background: Water polo (WP) is a high-intensity team sport that requires a combination of physical endurance, muscular strength, speed, and specific technical skills. Due to the demanding and prolonged nature of this sport, adequate and balanced nutrition plays a fundamental role in athletes’ performance, recovery, and overall health maintenance. Objectives: We aimed to compile all available information on the importance of ergonutrition and supplementation in the recovery of WP players. This will help in understanding this sport’s specific challenges and requirements, enabling players and coaches to design more effective recovery plans to optimize performance, achieve goals, and successfully cope with intense training and competition. Method: English-language publications were searched in databases such as Web of Science, Scopus, SciELO Citation Index, Medline (PubMed), KCI Korean Journal Database, and Current Contents Connect using a series of keywords such as WP, nutrition, recovery, and ergogenic aids individually or in combination. Results: In the field of ergonutritional recovery in WP, certain supplements such as whey protein, beta-alanine, L-arginine, spirulina, and copper can be beneficial for improving performance and recovery. In some cases, WP athletes may consider using ergogenic supplements to further improve their performance and recovery process. However, it is important to bear in mind that any supplement should be carefully evaluated under the supervision of a health professional or a sports nutritionist, as some supplements may present side effects or unwanted interactions. Conclusions: Adequate ergogenic nutrition adapted to the needs of WP players is essential not only to optimize their athletic performance but also to ensure effective recovery and maintain their long-term health and general well-being. The application of these strategies should be evidence-based and tailored to the individual needs of the players and the specific demands of the sport. Future experimental research that can confirm our results is essential.

## 1. Introduction

Water polo (WP) is an intermittent, high-intensity team sport that combines short, high-speed actions (movement, jumping, fighting, and shooting) with incomplete recovery periods and a game density with a work/recovery time ratio of 5:2 [1,2]. WP players require the simultaneous use of aerobic and anaerobic energy metabolism during competition [3]. In particular, there is a high demand in the aerobic system for the replenishment of phosphocreatine, the clearance of blood lactate concentration from muscle, and the elimination of accumulated intracellular inorganic phosphate [4].

Creatine is a popular sports and exercise supplement known for its positive effects on physical performance, especially during high-intensity, short-duration activities. There are some specific reasons why men may experience more or different benefits compared to women [5]. In general, men tend to have greater muscle mass than women due to hormonal differences (e.g., higher testosterone levels). Creatine is particularly effective in increasing muscle strength and size, which may be more pronounced in men due to their greater ability to gain muscle mass by helping to increase the production of ATP (adenosine triphosphate), the main source of energy for rapid muscle contractions. As men tend to have more type II muscle fibres (fast fibres), they can benefit more from creatine supplementation during explosive and high-intensity exercise. On a hormonal level, men have higher levels of testosterone, a hormone that plays a crucial role in muscle development and recovery. The interaction between creatine and testosterone can further enhance anabolic effects in men, who may also experience faster and more significant adaptations to resistance training when using creatine, resulting in more noticeable increases in muscle strength and size [6]. As men tend to store more creatine in their muscles due to their greater muscle mass, they may experience a more significant increase in intramuscular creatine levels after supplementation, which translates into greater ergogenic benefits. In terms of recovery, creatine may also help to improve recovery between sets during intense training. As men tend to train with heavier loads or greater volumes, this benefit may be particularly relevant to them.

Sodium bicarbonate (NaHCO_3_) is a compound that has been studied for its ergogenic effects, especially in high-intensity, short-duration sports. Its use as a supplement can offer benefits to both men and women, but there are some specific considerations that can make its use particularly relevant for women in mechanisms of action such as lactic acid buffering or increasing blood pH, since in general, women tend to have less muscle mass than men, which can influence the production of lactic acid during exercise [7]. The use of bicarbonate may be especially useful for women by helping to manage lactic acid build-up and improve their ability to perform intense exercise. Women may experience differences in recovery after exercise due to hormonal and physiological factors. The use of bicarbonate could help reduce muscle fatigue and hasten recovery after intense workouts. Some research also suggests that women may benefit from using bicarbonate during certain phases of the menstrual cycle when fatigue can be more pronounced [8]. In addition, women may have a different perception of physical exertion than men. The use of bicarbonate could help to reduce this perception during intense exercise, allowing for better performance.

It is precisely aerobic metabolism that dominates the energy supply, although the most decisive parts of the game depend on anaerobic metabolism [1]. However, physiological demands in WP are conditioned by multiple variables such as the proportion of different technical–tactical actions in each game, the strategies (game systems or match plans, for example) of each coach, gender, and level of performance. Differences in aerobic and anaerobic potential have been observed between international and national WP players [1], but not among the different playing positions.

In team sports such as WP, players face numerous challenges daily in training and competitions before fully recovering [9,10,11]. Improper recovery is associated with high levels of fatigue, decreased performance, and increased risk of injury [12,13]. Therefore, we can consider that recovery in sport is an essential parameter for performance, both at a general level [9,14] and in the discipline that concerns us [11,15]. In this sense, nutrition plays a key role in all regeneration processes, in performance and injury recovery [14,16,17], and in the maintenance of health and the prevention of diseases [18]. It has been demonstrated that the benefits of good nutrition are positively related to physical condition [19], cognitive skills [20], and mental health [21]. In sport, the impact of nutrition on performance has been the subject of research for the last decades. Athletes who follow appropriate nutritional strategies show higher performance in team and individual sports [22] in both men [23] and women [24]. Nutritional strategies play a supporting role in improving adaptation to training [17], and the timing of nutrient intake is related to the concept of optimal energy fuel [14].

WP is a team sport that combines anaerobic and aerobic elements, which makes it unique compared to other team sports. In terms of similarities, both WP and other team sports (such as football, basketball, or rugby) require a high level of physical fitness. Athletes must have stamina, strength, and agility to compete effectively. All these sports involve teamwork, communication, and strategy. Team cohesion and tactical understanding are essential for success. Athletes in all these sports require an adequate caloric intake to meet their energy requirements. This includes a combination of CHO, protein, and fat to optimise performance and recovery. As with other contact or high-intensity sports, WP players are at risk of injury, highlighting the importance of nutrition for recovery and overall health.

One difference between WP and other team sports is that it is played in an aquatic environment, which introduces unique factors such as water resistance and the need for specific swimming skills. This can affect both training and nutritional strategies. WP matches tend to consist of short, intense periods (20–30 s) followed by short rest periods, whereas other sports may involve longer periods of continuous activity (such as football) or alternate between intense and less intense efforts (such as basketball). Although both types of sport require anaerobic and aerobic energy, the specific exercise patterns of WP may require a different approach to macronutrients. For example, a greater emphasis on carbohydrate intake may be required to maintain energy levels during short, intense intervals. The literature emphasises the importance of personalising nutritional strategies according to the individual needs of the athlete and the specific demands of the sport. This applies to WP as well as other sports.

From this point of view, nutrition and ergogenics are fundamental in the recovery process of WP players [11]. Their importance lies in the fact that they optimize energy replenishment and improve the recovery capacity of these athletes [25]. On the one hand, adequate nutrition, which sometimes includes strategic supplementation, guarantees a sufficient supply of essential nutrients for energy synthesis and the repair of tissues damaged during sports practice and recovery [26]. A diet high in protein and moderate in carbohydrates prioritising foods with a high nutritional density may be appropriate throughout the training cycle [27]. However, dietary needs vary during the season, so they should be organised into periods and individualised [27,28].

On the other hand, ergogenic strategies go beyond simple nutrition and include specific training techniques that improve overall physical performance [29]. These training programs seek to increase the aerobic and anaerobic capacity of athletes, which in turn optimizes energy use during competition and facilitates more efficient recovery. By combining optimal nutrition with these ergogenic strategies, water polo athletes can maximize results in terms of recovery and long-term performance [25].

For these reasons, one of the most commonly used recovery strategies in sports is ergonutritional support [9,30]. These supplements should be used with caution and as part of a general and individualized overall nutrition and performance plan [29] supervised by an expert [27,28]. In fact, before starting any supplementation, the athlete should already be following a diet that is sufficiently rich and balanced in proteins, carbohydrates, fats, and micronutrients so as not to limit its additional benefits [9]. Improvements in the health and performance of athletes have been demonstrated using, among others, the following ergonutritional aids: caffeine [31], creatine [32], sodium bicarbonate (NaHCO_3_) [7], nitrates [33], β-alanine [34], and/or ashwagandha [35]. Different types of reviews on this subject have been carried out in sports such as basketball [36], volleyball [37], tennis [38], and swimming [39]. However, to the authors’ knowledge, no previous review on WP has been published. Therefore, the main objectives of this systematic review were: (i) to organise nutritional strategies; (ii) to evaluate the effect of ergonutritional aids in WP; to help coaches and improve player performance; and (iii) to establish which of these ergonutritional aids are effective for recovery.

## 2. Methods

### 2.1. Sources of Information

This article provides a systematic evaluation of nutritional and ergogenic strategies in water polo recovery. The review was carried out following the guidelines established in the Preferred Reporting Items for Systematic Review and Meta-Analysis (PRISMA) [40].

The PICOS model was applied to define the inclusion criteria described in [41] (Figure 1).

This review focused on studies that met certain inclusion criteria: (I) the study population had to be composed of WP players, (II) the articles had to examine aspects such as ergonutritional supplementation and associated recovery strategies, as well as ergogenic aids in WP, and (III) the study designs had to be non-randomised. Relevant studies were selected for review, applying exclusion criteria: (i) studies with participants from other disciplines or with previous disorders (ÁM-O and JB-G), (ii) articles on other sports populations, abstracts, non-peer-reviewed articles, and book chapters (ÁM-O and JB-G), and (iii) studies unrelated to nutritional needs, recovery strategies, or ergogenic aids (ÁM-O and JB-G).

Author ÁM-O carried out exhaustive structured electronic searches on several databases of scientific literature, such as Web of Science (WOS), SciELO Citation Index, Medline (PubMed), Current Contents Connect, KCI Korean Journal Database, and Scopus (Figure 2), using a combination of keywords related to WP, nutrition, recovery and ergogenics: ((‘water polo’ [MeSH Terms] OR ‘water polo’ [All Fields]) AND (‘nutrition’ [MeSH Terms] OR ‘nutrition’ [All Fields]) AND (‘recovery’ [MeSH Terms] OR ‘recovery’ [All Fields]) OR (‘ergogenic’ [MeSH Terms] OR ‘ergogenic’ [All Fields])). These keywords were selected based on the opinion of the authors (ÁM-O and JB-G), the review of the literature, and controlled vocabulary such as Medical Subject Headings (MeSH). The search was not limited by publication date but was restricted to human studies written in English. The last search was conducted on 3 January 2025. In addition, the references of the retrieved articles were examined to analyse and discuss the most relevant studies in this field using the snowball strategy [42]. This study was registered in the International Prospective Register of Systematic Reviews (PROSPERO) on 17 January 2025 (registration number: CRD42025635021).

### 2.2. Study Selection

The process of identifying and selecting the relevant studies for this systematic review was conducted in several meticulous stages. First, an exhaustive search was conducted on the main bibliographic databases using the above search strategy (ÁM-O and JB-G). This allowed us to retrieve an initial set of potentially relevant publications [36], whose titles and abstracts were carefully examined by two of the authors (ÁM-O and JB-G) to identify duplicates and discard those that did not meet the established eligibility criteria. In cases of disagreement, the option was taken to include a third author (EG-O) to clarify the choice. Subsequently, the full texts of all the studies that were classified as relevant in the previous stage were obtained. These articles were reviewed in depth by the same authors (ÁM-O and JB-G), who rigorously applied the predefined inclusion and exclusion criteria to finally select the trials that would be included in this systematic review. During this process, special attention was paid to the reference section of each of the relevant articles to identify possible additional studies that might be eligible (ÁM-O).

### 2.3. Data Extraction

Relevant data were extracted from each study (ÁM-O and JB-G), such as the source, the population, the methods, the characteristics of the intervention, and the significant differences among groups. The included studies were then organised into two groups: (a) nutritional demands and (b) the relationships among these demands, recovery, and performance support. The authors discussed and reached a final consensus to minimise errors during data extraction and group formation (ÁM-O and JB-G).

The process of compiling and grouping data was not simple, as some studies did not have enough information or contained inconsistencies in their reports. However, through careful review and discussion among the authors, a full understanding of the relevant data was achieved. The studies included were selected according to rigorous criteria, such as the use of randomised controlled trials and the inclusion of a control group, which guaranteed the quality and reliability of the information. The process of data extraction and collation played a fundamental role for the authors of the analysis (ÁM-O and JB-G), as it allowed for the systematic examination of the data and the identification of important patterns and trends regarding the relationship between energy demands, quality control, nutrition, and performance.

### 2.4. Assessing the Quality of Experiments: Risk of Bias and Levels of Evidence

To obtain reliable conclusions, the STROBE^®^ (Strengthening the Reporting of Observational Studies in Epidemiology) guidelines were used to assess the quality of the publications following the guidelines of the Cochrane Collaboration [44], and to carefully consider the possible limitations of the included studies, the STROBE^®^ guidelines for reporting observational studies [45] (Figure 3) were used to assess the quality of the publications. The guidelines to be evaluated are as follows.

*Random sequence generation:* This process involves the random allocation of participants to the different groups in the study (for example, the treatment group and control group). The main objective is to ensure that the allocation is not influenced by the researcher or by the characteristics of the participants, eliminating selection bias. An appropriate method for generating random sequences is essential to establish initial comparability between the groups. Inappropriate methods, such as alternation or allocation based on dates of birth, can introduce bias.

*Allocation concealment:* Once the random sequence has been generated, it is imperative to conceal the allocation of each participant until the precise moment of their inclusion in the study. This prevents researchers or staff from influencing the allocation based on their knowledge of which group a participant is expected to belong to. Concealing the allocation protects the random sequence from manipulation, guaranteeing impartiality in the allocation of participants. Effective methods include the use of sealed and sequentially numbered envelopes, identical containers for the treatments, or centralised allocation through an automated computer system.

*Blinding of participants and personnel:* Blinding seeks to prevent knowledge of the assigned treatment from influencing the behaviour of the study participants and staff. Double-blinding, where neither the participants nor the staff administering the treatment know which group anyone belongs to, is the gold standard. Blinding can be difficult to implement in some studies (for example, in surgical or behavioural interventions), but it is crucial to explore alternative strategies to minimise bias, such as single-blinding (only the participants are unaware) or blinding of outcome assessors. The placebo effect, modification of behaviour by staff due to expectations, and differences between groups in terms of attention received are examples of biases that masking seeks to mitigate. But even if participants and staff are not masked, those who evaluate the results (for example, doctors who measure blood pressure, and researchers who analyse survey data) must not be aware of the treatment assignment. This masking reduces the risk of detection bias, where knowledge of the assigned treatment influences how the results are evaluated.

*Blinding of outcome assessment:* Also known as ‘detection bias’ when referring specifically to outcomes reported directly by patients, this represents a significant threat to the validity of clinical research. It refers to a situation where knowledge of the assigned intervention (whether active or placebo) influences the way participants or researchers evaluate or report results. In other words, the expectation or belief about the intervention can bias the patient’s perception or the evaluator’s interpretation, leading to results that do not reflect the true effectiveness of the treatment.

This bias is especially problematic when the results are based on information provided directly by patients, as their subjective perception is more susceptible to the influence of their expectations. For example, if a patient knows that they are receiving a promising new treatment for pain, they may be more likely to report a decrease in pain, even if the treatment does not have a real physiological effect. Conversely, a patient who believes they are receiving a placebo may be more likely to minimise any improvement they experience.

To mitigate detection bias in studies that use patient-reported outcomes, it is crucial to implement rigorous masking strategies. ‘Masking’ (or blinding) refers to the process of concealing treatment allocation from both patients and the researchers who evaluate the results. Ideally, both participants and evaluators should be ‘blind’ to the intervention they are receiving or administering.

However, perfect blinding is not always possible, especially in complex interventions or when there are distinctive side effects associated with the active treatment.

*Incomplete outcome data:* The analyses carried out have significant limitations due to the presence of incomplete outcome data, which may affect the validity and generalisation of the conclusions. There is an attrition bias, characterised by the loss of participants throughout the study, especially in the long-term follow-up (considered to be periods longer than 6 weeks). This selective loss of participants can introduce a systematic bias if those who left the study differ significantly from those who remained, for example, in terms of the severity of the condition, response to treatment, or side effects experienced. The absence of data from these participants can lead to an overestimation or underestimation of the observed effects.

*Selective reporting:* This refers to the tendency to report certain results preferentially, whether by researchers, study sponsors, or even the participants themselves. This bias can manifest itself in the omission of negative or non-significant results, the selective presentation of statistical analyses that favour certain conclusions, or the exaggeration of beneficial effects. Selective reporting distorts the available evidence and hinders the objective evaluation of the true effect of the treatment or intervention studied. It is crucial to take these limitations into account when interpreting the results and to consider the need for additional research that addresses these potential biases.

## 3. Results

To ensure the rigour and validity of the results of our review, we determined the level of evidence of each of the selected studies. For this, we used the Oxford quality scoring system (Table 1), a method that is widely recognised and used worldwide [46,47] for its robustness and clarity. This tool, specifically designed to assess the quality of clinical trials, has been established as an international standard of reference [48] in medical research. Its widespread use is due to its ability to assess the methodological quality of clinical trials in an objective and standardised manner, thereby minimising Rater bias [49]. By providing a structured and transparent assessment, this system greatly facilitates the individual interpretation of each study, and more importantly, the comparison of the results of different studies [50]. This aspect is particularly critical in the context of systematic reviews and meta-analyses, where a thorough and systematic assessment of the quality of the included studies is essential to ensure the validity and reliability of the overall conclusions drawn from the combined analysis of the available evidence [51]. Without rigorous quality assessment, the conclusions of a systematic review could be compromised by the inclusion of studies with flawed methodologies, which could lead to erroneous conclusions and inappropriate clinical decisions [52]. Therefore, the choice of the Oxford scoring system underlines our commitment to transparency and objectivity in the synthesis of scientific evidence.

An initial search of the scientific literature identified 10,438 articles related to WP and other keywords for inclusion in this systematic review. Of these, 7624 articles unrelated to nutrition, recovery, or performance aids were excluded. A further 1142 dealing with other disciplines (not meeting the inclusion criteria) were also eliminated (Figure 3).

To identify studies that were not related to the specific topic under review, we defined the inclusion and exclusion criteria above and conducted an exhaustive search of the academic databases listed above, using the keywords defined in the relevant section.

Once the search results were obtained, an initial review was performed by evaluating the titles and abstracts to identify studies that met the defined criteria. Studies that were not relevant were excluded at this stage. The studies that passed the initial screening were reviewed in their entirety. The full content of each study was examined to determine whether it addressed the topic of interest according to the defined criteria. If during this review, it was determined that a study did not meet the criteria (for example, it did not include the specific intervention or target population), it was excluded. Finally, after screening and full review, the relevant studies that met all the established criteria were compiled and a critical analysis of the methodological quality and relevance of these selected studies was carried out.

After an exhaustive review of the available scientific literature and rigorously applying the previously defined inclusion criteria, which included the requirement that the studies focus exclusively on water polo, a limited number of relevant works were identified. Specifically, a total of eleven articles (Table 2) were found that met the established requirements and were therefore incorporated into the analysis. This figure, unfortunately, highlights the notorious scarcity of scientific research specifically dedicated to water polo, suggesting a prevailing need to foster and promote more in-depth and extensive studies on this sporting discipline. The limited amount of existing literature restricts the knowledge base available to coaches, players, and researchers, potentially limiting the development and optimisation of performance in this sport.

The age of the WP players in the studies ranged from adolescent to adult for both sexes. One study was published in 2024 [62], one in 2022 [61], one in 2021 [60], one in 2020 [59], two in 2018 [57,58], one in 2017 [56], two in 2016 [54,55], one in 2014 [25], and one in 2010 [53], which means that only four of them were published in the last five years. In addition, eight studies were published in a journal with an index factor of Q1 [25,53,55,56,57,58,60,62], two in Q2 [54,59], and one in Q3 [61].

### 3.1. Assessing the Quality of Experiments: Risk of Bias and Levels of Evidence

To evaluate the general risk of bias for each category (Figure 3), two authors (AM-O and JB-G) independently assessed the methodological quality and risk of bias of the studies. Any disagreement was resolved by a third-party assessment (EG-O), following the guidelines of the Cochrane Collaboration [44]. A scale was used to classify the quality of the studies: (1) good quality (>14 points, low risk of major or minor bias); (2) fair quality (7–4 points, moderate risk of major bias); and (3) poor quality (<7 points, high risk of major bias) [45].

The distinct aspects reviewed were classified into several domains: random sequence generation (selection bias), allocation concealment (selection bias), blinding of participants (implementation bias), blinding of outcome assessment (detection bias), incomplete outcome data (attrition bias), selective reporting (reporting bias), and other types of biases. They were categorised as ‘low’ if they met the criteria for negligible risk of bias (plausible bias unlikely to seriously alter the results) or ‘high’ if they met the criteria for substantial risk of bias (plausible bias that seriously undermines confidence in the results). If the risk of bias was unknown, it was considered ‘unclear’ (plausible bias that raises some doubt about the results) [44].

Most of the reviewed studies lacked clear criteria for other types of biases, due to incomplete reporting or omission of relevant variables, such as caloric intake or dietary habits. The robustness and practical applicability of a study’s ergonomic and nutritional recommendations were assessed using a tool designed to evaluate nutrition and performance research [63] (Figure 3 and Figure 4).

### 3.2. Nutritional Strategies

Nutritional strategies can improve daily training, recovery, and performance on match day [25]. These same authors [25] emphasised that daily carbohydrate needs are likely to be 4–8 g/kg body mass per day and that consuming 20–30 g of protein immediately after training is sufficient to maximise muscle building [25]. However, a much lower protein and carbohydrate intake than international recommendations was observed in nineteen elite Hungarian WP players, and a protein and carbohydrate intake well below international recommendations was observed during a four-month intervention [61]. Consequently, adequate planning and education must be provided to ensure that nutritional requirements [25] are met. Therefore, to ensure that the dietary needs of WP players are fully met, it is essential to implement thorough planning and effective education programmes [64].

### 3.3. Recovery

Among the supplements used in recovery, it can be seen that products such as whey protein, β-alanine [65], L-arginine [66], and spirulina plus copper are suitable for improving performance and recovery. These supplements have contrasting functions and benefits for athletes. Whey protein is a high-quality source of protein that helps repair and build muscle after training. β-alanine increases carnosine levels in the muscle, which improves endurance capacity. L-arginine is an amino acid that participates in the production of nitric oxide, improving blood flow and muscle oxygenation. Spirulina, together with copper, is a nutrient-rich supplement that helps fight oxidative stress and fatigue.

On the other hand, nitrates work adequately to improve performance in underwater apnoea activities, as they increase the availability of nitric oxide and improve the efficiency of oxygen use [67]. As for other supplements, it has been seen that HMB [68] has no influence on strength and that NaHCO_3_ is beneficial in women, but that creatine has no clear benefit in men. This shows that the effectiveness of supplements can vary according to sex and type of physical activity.

### 3.4. Ergogenic Supplementation

In decreasing order, the most evaluated supplements were β-alanine [55,56,58], L-arginine [60], spirulina [62], phosphorus [59], sour cherry juice [54], NaHCO_3_ [53], and beet juice [57].

Several authors investigated the effect of four weeks of β-alanine supplementation on repeated sprint capacity [55] and on VO2 and strength associated with VO_2pmax_ or peak [58] with twenty-two elite players from Brazil. Participants received 4.8 g·day^−1^ of the supplement (placebo or β-alanine) for the first ten days and 6.4 g·day^−1^ for the last eighteen days, with a total of 163.2 g over twenty-eight days. Four weeks of β-alanine supplementation slightly improved the ability to perform repeated sprints during the first quarter of a WP match, but not necessarily in subsequent periods [55], as well as the strength associated with VO2pmax [58]. An improvement in performance (shooting speed) with a six-week β-alanine supplementation has also been highlighted [56] in young players.

Likewise, four weeks of ingesting 5 g·day^−1^ of L-arginine increased oxidative metabolism during exercise in elite players from Italy [60], and eight weeks with a dietary supplement containing spirulina (titrated in phycocyanin 1 mg/mL) and copper improved subjective measures of performance and reduced muscle stress [62]. On the other hand, it has been observed that the ingestion of 100 g of glucose with phosphorus in senior players did not improve performance [59]. This aspect was not improved with ingestion of sour cherry juice for six days in elite players [54], ingestion of NaHCO_3_ (0.3 g/kg) in international female elite players from Australia [53], or supplementation for six days with beetroot juice rich in nitrates (~800 mg·day^−1^ nitrate) in female elite international players from the Netherlands [57].

### 3.5. Effect of Nitrates and β-Alanine: A Narrative Meta-Analysis

The dependent variables found most ‘repeatedly’ (nitrates 2; β-alanine 3) within the works used that met the inclusion and exclusion criteria (11) show that research in this discipline is neither broad nor extensive.

The optimisation of performance in this sport may benefit significantly from supplementation with nitrates and beta-alanine, two compounds that have been the subject of research in general sport. This narrative meta-analysis aims to review the existing literature on the effects of these supplements on physical performance and recovery in WP players.

The mechanism of action is that nitrates, found in foods such as beetroot, spinach and other green leafy vegetables, are converted in the body to nitric oxide (NO), which acts as a vasodilator, improving blood flow and oxygen delivery to the muscles during exercise [69]. Several studies have shown that nitrate supplementation can increase the efficiency of oxygen use during aerobic exercise [70,71,72]. This is important for WP players, who require good aerobic capacity to maintain their performance during prolonged matches, although a study by Tan et al. [53] suggests that the effects of NaHCO_3_ intake on average sprint performance were not substantial. These results contradict other studies on NaHCO_3_ and team sports performance [73,74], but this study is unique in that it examined highly trained WP players performing sport-specific tasks. The current advice for WP players is not to expect a significant improvement in intermittent sprint performance with NaHCO_3_ supplementation. On the other hand, Jonvik et al. [57] suggest that the effects of beetroot juice supplementation do not improve intermittent sprint performance in elite WP players, but there may be potential beneficial effects during dynamic apnoea. On another note, nitrates may reduce muscle fatigue by improving blood flow and helping to reduce the accumulation of metabolites that cause muscle fatigue, allowing athletes to maintain high levels of performance for longer.

Although research has mainly focused on their effect on aerobic performance, some studies suggest that nitrates may also benefit short, intense anaerobic efforts, which are crucial in a sport such as powerlifting.

Recent research has shown that nitrate intake can significantly improve performance in intermittent sports. A study by Domínguez et al. (2018) [75] found that athletes who consumed beetroot juice before exercise showed improvements in their ability to perform repeated, high-intensity activity.

As for β-alanine, its mechanisms of action cause it to combine with histidine to form carnosine, a dipeptide that acts as a lactic acid buffer in muscles. This helps to delay muscle fatigue during intense exercise. Beta-alanine has been shown to be effective in improving performance during anaerobic and intermittent exercise [34], which is essential for WP players, who perform explosive movements. Supplementation with β-alanine can allow athletes to work more before fatigue, which is crucial in competition where every second counts [65]. By reducing the build-up of lactic acid, β-alanine may facilitate faster recovery between sets or intense periods of play. A meta-analysis by Hobson et al. (2012) [76] concluded that β-alanine supplementation significantly improved performance during intermittent and prolonged exercise.

Both supplements have complementary benefits for WP players. Nitrates focus mainly on improving aerobic performance and reducing muscle fatigue by improving blood flow and oxygen delivery. Beta-alanine focuses on improving anaerobic performance and increasing the capacity for prolonged intense effort by buffering muscle acidosis. The strategic combination of both supplements could provide a comprehensive approach to optimising both performance and recovery in WP players. However, it is important to consider individual factors such as specific nutritional needs, dietary preferences, and individual responses to each supplement.

This narrative meta-analysis provides an overview of how nitrates and β-alanine can be used strategically by WP athletes to improve their athletic performance and facilitate effective recovery after intense exercise.

## 4. Discussion

The main aims of this systematic review were: (i) to organise nutritional strategies; (ii) to evaluate the effect of ergonutritional aids in WP and assist coaches in applying evidence-based strategies for enhancing player performance; and (iii) to establish which of these ergonutritional aids are effective for recovery.

The results obtained were analysed in comparison with previous studies to contextualise the findings in the scientific and sporting fields. In the field of ergonutritional recovery in WP, certain supplements such as whey protein, beta-alanine, L-arginine, spirulina, and copper can be beneficial for improving performance and recovery.

The under-representation of female players in water polo research is a significant problem that has important implications for the validity, applicability, and relevance of evidence-based recommendations for this sport. This often-unnoticed disparity undermines the ability to provide training, nutritional and injury prevention strategies that are truly personalised and optimised for female athletes. When research is based on male data, conclusions cannot be safely extrapolated to women, as there are inherent physiological, hormonal, and biomechanical differences that affect their performance and susceptibility to injury.

To effectively address this gap and strengthen the knowledge base on women’s water polo, it is imperative that researchers take an initiative-taking and deliberate approach. Firstly, future study designs should explicitly prioritise the inclusion of women by establishing clear and transparent inclusion criteria to ensure equitable representation. This means not only recruiting female participants but also ensuring an adequate sample size to produce statistically significant and robust results. In many cases, this will require a collaborative and sustained effort with women’s water polo clubs, sports federations, and other relevant organisations to facilitate the recruitment of players and encourage their active participation in research. It is essential to clearly communicate the objectives of the research, the potential benefits to the players, and to ensure the confidentiality and anonymity of the data collected.

Furthermore, future research should go beyond simply including women in existing studies. Research should be conducted that focuses exclusively on women and addresses crucial aspects such as gender-specific physiology, the psychology of female performance, the specific nutritional needs of female water polo players, and the impact of hormonal factors on their performance. This includes, for example, detailed studies of the influence of the menstrual cycle on strength, endurance, and recovery, as well as research into the prevalence and mechanisms of gender-specific injuries.

It is also essential to conduct systematic comparative analyses between men and women to identify key differences in performance, injuries, and training needs. These analyses should go beyond simple comparisons of averages and explore the interactions among biological, psychological, and social factors that contribute to the observed differences. For example, gender differences in throwing biomechanics, body composition, aerobic and anaerobic capacity, and response to training could be investigated.

Finally, research on women’s water polo needs to adopt methodologies that consider the specificities of women’s sport. This means using assessment and training protocols that are sensitive to hormonal fluctuations.

Assorted studies have analysed the physiological requirements and energy demands of elite WP players [1]. These works have allowed for a better understanding of the characteristics of this sport and the players’ needs from a scientific perspective. In this context, the present discussion analyses the results obtained in comparison with previous studies to contextualise the findings in the scientific and sports fields.

### 4.1. Nutrition

The findings of numerous studies show that a sizeable proportion of elite players do not meet international recommendations for daily CHO or protein intake, which can hurt their performance and recovery [14,22,77,78]. This coincides with the significant deficiency observed [61] in the diet of elite Hungarian players, with an insufficient intake of key micronutrients.

This problem is widely documented in team sports, where a lack of nutritional education and professional advice negatively affects the planning and implementation of a suitable diet [25]. Many high-level athletes do not have the necessary knowledge about the specific nutritional requirements of their sport or their position within the team, which leads them to adopt sub-optimal eating habits [22].

Furthermore, the dynamics and demands of team sports, with intense competition and training schedules, can make it even more difficult to adhere to a balanced diet with sufficient essential nutrients [29]. Players often face logistical and time challenges that prevent them from planning and eating proper meals, exacerbating macronutrient intake problems [22].

The results regarding carbohydrate intake suggest an average intake below the figure recommended by other previous studies [25,79,80,81,82,83]. These articles suggest that an intake of 4–8 g/kg/day of carbohydrates is optimal for improving performance in sports such as WP. This lower carbohydrate intake could hurt the athletes’ recovery capacity and compromise their energy demands during training and competition. Therefore, future lines of research should be focus on this in WP players.

In addition, it was observed that protein intake strategies do not always conform to the recommendations of 20–30 g of protein in post-workout meals, as is already known [14]. These researchers point out that this protein intake is key to stimulating muscle synthesis and promoting adequate recovery in athletes.

It is important to emphasise that although players perceive the importance of nutrition, they do not always implement the recommendations. These results are in line with the findings of [84], who concluded that WP players need specific training to clear up misconceptions and improve the practical application of nutritional strategies. The evidence reinforces the need to continually customise and educate athletes and coaches on the basic principles of nutritional periodization.

Periodised nutrition is an approach that adapts nutrient and calorie intake to the specific demands of training and phases of the sporting season [17]. This approach can offer several benefits, such as optimising performance by adjusting calorie and macronutrient intake according to training needs, allowing athletes to maximise energy and performance at key moments such as competition or intense training. Periodised nutrition allows athletes to manipulate their intake to promote fat loss or muscle gain at specific times, helping to achieve aesthetic or functional goals [29]. More efficient recovery by adjusting CHO and protein intake according to the type and intensity of exercise can improve muscle recovery and reduce the risk of injury. Proper nutrition can help maintain a strong immune system and prevent injuries associated with overtraining. Periodised nutrition also allows for some flexibility in diet, which can make it more sustainable for athletes in the long term by avoiding extreme restrictions [29].

On the other hand, periodised nutrition offers a strategic approach that allows athletes to adapt to changing physical demands throughout the year [85]. By carefully adjusting caloric intake and macronutrients according to training and competition phases, athletes can optimise performance, improve body composition and facilitate effective recovery.

### 4.2. Recovery

Recovery in high-intensity sports such as WP is essential to mitigate accumulated fatigue and avoid injury [11]. Athletes who practise these disciplines face remarkably high training and competition loads, which can lead to physical and mental exhaustion if recovery processes are not effectively managed [86].

The results of numerous studies indicate that although players recognise the importance of recovery strategies at the ergonutritional level, their application is often unconscious or sporadic. Many athletes tend to prioritise training and competition, relegating recovery techniques to the background. This attitude is due in part to the fact that they are not always aware of the benefits that these methods provide in the short and long term.

The existing literature emphasises that nutritional recovery strategies, such as glycogen replacement and antioxidant consumption, are essential after intense efforts [87]. Several studies have investigated the use of different foods and supplements to improve recovery after intense physical activities [88]. For example, the use of sour cherry juice as a recovery aid in WP players was evaluated [54]. The researchers found that while sour cherry juice may have beneficial effects in some sports, its effectiveness is limited in the specific case of WP players. This underlines the importance of selecting recovery strategies based on scientific evidence specific to each sport [89].

Furthermore, it has already been emphasised [9] that recovery needs vary significantly among players depending on their position and role in the team, a point that is also observed in the present research.

### 4.3. Ergogenic Supplements: Scientific Evidence

Supplementation in sport is a topic of great interest and debate, with many athletes turning to various supplements to improve their performance, hasten recovery, and optimise their overall health. It is well known that a large proportion of athletes, both professional and amateur, use some form of supplement in the hope of improving physical performance, increasing strength or endurance, reducing fatigue during exercise, or speeding up recovery after intense exercise. In this case, we stick to the data collected in the studies included in the paper.

Ergogenic nutritional supplementation is presented as a potential and promising tool for improving performance in water sports such as WP. However, the results of this comprehensive review reveal mixed findings regarding the effectiveness of different ergogenic aids, specifically in this sport.

Beta-alanine supplementation has become an area of considerable interest in sports research, particularly in disciplines characterised by intermittent high-intensity efforts. Several studies have investigated its effects on athletic performance, shedding light on its potential benefits. A prominent example is a study by Brisola et al. (2016) [55], which investigated the effects of beta-alanine supplementation in water polo players. The results of this study showed that four weeks of beta-alanine supplementation resulted in a significant improvement in repeated sprint ability (RSA), a key measure of performance in this sport. Importantly, this improvement was particularly evident in the first quarter of matches, suggesting that beta-alanine may be particularly useful in maintaining explosiveness and speed at the start of competition, when fatigue has not yet built up significantly. In a similar line of research, Claus et al. (2017) [56] investigated the effects of beta-alanine supplementation on shooting speed, a fundamental skill in many sports. Their results showed that six weeks of beta-alanine supplementation resulted in significant improvements in shooting speed. This finding suggests that beta-alanine not only benefits resistance to fatigue but could also improve the power and speed of explosive movements.

These findings from independent studies provide converging evidence of the potential benefits of beta-alanine for athletes involved in intermittent sports. The present study is consistent with these observations and confirms the hypothesis that beta-alanine has a positive effect on physical performance. Specifically, the results indicate that beta-alanine improves muscle buffering capacity, a critical mechanism for reducing the accumulation of lactic acid and other metabolites that contribute to fatigue. By increasing muscle buffering capacity, beta-alanine helps delay the onset of fatigue during high-intensity exercise, allowing athletes to maintain optimal performance for longer periods of time. Evidence suggests that beta-alanine may be a valuable tool for improving performance in sports that require intermittent high-intensity effort by improving repeated sprint capacity, shooting speed, and delaying muscle fatigue.

L-arginine, a non-essential amino acid, has gained considerable attention in the field of sports nutrition and physical performance due to its crucial role in the synthesis of nitric oxide (NO). Nitric oxide acts as a powerful vasodilator, relaxing the walls of blood vessels and facilitating blood flow. This improvement in blood flow results in more efficient delivery of oxygen and nutrients to the muscles during exercise, which can potentially improve performance. In a related study, Gambardella et al. (2021) [60] investigated the effects of L-arginine supplementation in elite water polo players. The researchers observed a significant increase in oxidative metabolism in these athletes after four weeks of supplementation. This finding suggests that L-arginine may improve the body’s ability to use oxygen during exercise, which is critical for endurance performance. The findings in the literature are consistent with and support the observations of Gambardella et al. (2021) [60] and strengthen the hypothesis that L-arginine may offer significant benefits for endurance performance. Specifically, we propose that this amino acid may be particularly valuable during prolonged periods of play or training, when muscular oxygen demand is high, and when fatigue may be a limiting factor. By improving oxygen delivery, L-arginine could help athletes maintain optimal performance for longer periods of time.

Despite these promising results, it is important to recognise that research in this area is ongoing. Further studies are needed to fully translate the potential of L-arginine into effective supplementation strategies. It is important to determine the optimal dose of L-arginine for different types of athletes and physical activities. More research is also needed to understand the long-term effects of L-arginine supplementation, including possible side effects and the safety of long-term use. Future studies could also explore the interactions between L-arginine and other supplements or dietary interventions to maximise its benefits for athletic performance.

Creatine monohydrate is well established in strength and power sports, yet its efficacy in aquatic environments remains debated. Arazi et al. (2021) [32] noted improvements in short-duration, high-intensity activities, whereas Tan et al. (2010) [53] found no significant benefits in elite female water polo players. Our analysis suggests that creatine’s effectiveness may depend on individual variability and training status, highlighting the need for personalised supplementation strategies.

Caffeine has been extensively studied for its effects on endurance, cognitive function, and reaction time. Diaz-Lara et al. (2024) [31] demonstrated that caffeine consumption improved cognitive alertness and sprint performance in intermittent sports. Our review supports these findings, emphasizing caffeine’s potential benefits for maintaining focus and optimizing performance during prolonged matches. However, individual tolerance must be considered to avoid adverse effects such as increased heart rate and gastrointestinal discomfort.

Sodium bicarbonate (NaHCO_3_) has been shown to buffer acidosis, enhancing anaerobic performance. Grgic et al. (2020) [7] found significant benefits in repeated high-intensity efforts, particularly in female athletes. Our review corroborates these findings, suggesting that female water polo players may experience greater performance gains from NaHCO_3_ supplementation compared to males. However, gastrointestinal discomfort remains a limiting factor.

Nitrate supplementation, primarily through beetroot juice, has demonstrated efficacy in improving oxygen efficiency. Senefeld et al. (2020) [33] highlighted its potential for endurance-based activities. Jonvik et al. (2018) [57] explored its application in water polo, finding positive effects in dynamic apnoea and intermittent sprint performance. Our findings suggest that nitrates may be beneficial for underwater phases and prolonged exertion, though further research is required to optimize dosing protocols.

## 5. Strengths, Limitations, Future Research Lines, Practical Applications

The strength of the present review is that as it is an updated review, it includes the most recent studies, providing a modern and relevant view of the subject. This work can identify under-researched areas, which is crucial for guiding future research. A global survey involving 231 WP players and 76 coaches found a discrepancy between the perception of recovery strategies and the frequency of their use [15]. Specifically, 79 of the 231 players perceived nutritional supplements as useful recovery strategies, and 42 responded that they used them. Likewise, 29 of the 76 coaches considered nutritional supplements useful and 17 used them. In the case of ergogenic aids, eight players highlighted them as useful and five used them. Among the coaches, nine mentioned them as a useful strategy and three used them. Consequently, the lack of knowledge about recovery strategies among coaches could lead to incorrect models that increase the risk of overtraining [15].

It also provides a solid basis for designing interventions and educational programmes based on the latest evidence. A multidisciplinary approach to nutrition, recovery, and ergogenics allows for integrating knowledge from different fields, thus enriching the analysis.

However, the lack of recent studies may limit the amount of data available for analysis, which together with differences in study design may make it difficult to compare and synthesise results. On the other hand, most studies may have focused on specific populations, limiting the generalisability of the findings to different populations, and most studies may have been of short duration, preventing the observation of long-term effects.

On the other hand, the strength of the conclusions of a study or piece of research is intrinsically affected by the inherent limitations that result from the risk of bias. This risk, present at various stages of the research process, from sample selection to interpretation of results, can introduce systematic biases that affect the validity and generalisability of the conclusions. The greater the risk of bias, the less confidence can be placed in the conclusions reached, as they may reflect, in part or in full, the influence of factors unrelated to the variable or phenomenon under study. Therefore, a thorough assessment of the possible sources of bias and the rigorous implementation of strategies to mitigate its impact, as has been done, are crucial to ensure that the conclusions are robust and representative of the reality they are intended to describe. The significant presence of bias, if not adequately mitigated, undermines the credibility of the study and weakens the ability to draw valid conclusions applicable to wider populations.

This review, based on a careful analysis of eleven studies that comply with the established standards for this sport, shows that there are few specific data and little research on the subject. This lack of scientific studies indicates that it is an under-explored area that needs more research.

Due to the small number of studies reviewed, the conclusions should be taken with caution and cannot be applied to all people or situations. The results may not be valid for everyone due to the small sample, so the conclusions are only a first step and should be confirmed with further research using larger and more varied samples.

As a future line of research, it would be extremely interesting to conduct longitudinal studies to assess the long-term effects of nutritional and ergonomic interventions and investigate the effectiveness of personalised interventions based on athletes’ individual needs, develop education and training programmes for health and sports professionals based on the latest evidence, encourage collaboration among nutritionists, physicians, coaches, and other experts to design and evaluate comprehensive interventions to be applied, and to explore the use of new technologies, such as mobile applications and monitoring devices, to support athletes’ nutrition and recovery.

## 6. Conclusions

The organisation of nutritional strategies is a cornerstone in meeting the specific and inherently high energy demands of WP.

Among the many options available, β-alanine and L-arginine emerge as two supplements with a relatively strong evidence base. It is important to consider potential interactions with other supplements or medications, as well as individual side effects, before recommending their use.

Nutrition, recovery, and ergonomic support are fundamental pillars of the performance of WP players. The implementation of these strategies should be evidence-based and personalised according to the individual needs of the player and the specific demands of the sport.

## Figures and Tables

**Figure 1 nutrients-17-01319-f001:**
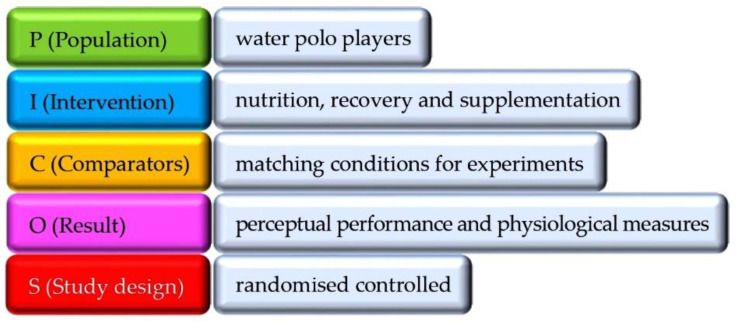
PICOS models.

**Figure 2 nutrients-17-01319-f002:**
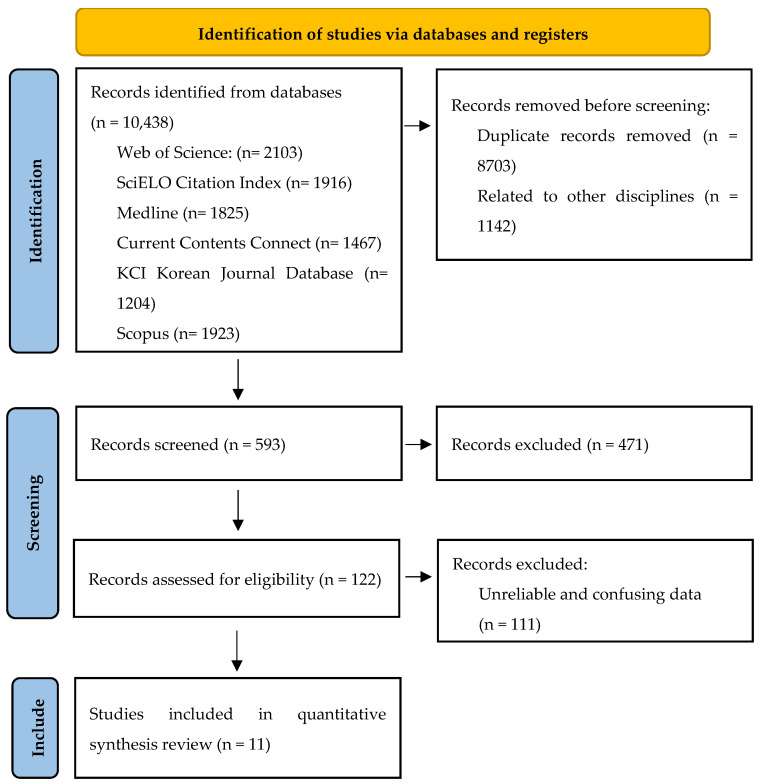
Study selection flowchart [43]. The PRISMA^®^ 2020 flow diagram templates are distributed by the terms of the Creative Commons Attribution (CC BY 4.0) license, which permits others to distribute, remix, adapt, and build upon this work, for commercial use, provided the original work is properly cited. To view a copy of this license, visit https://creativecommons.org/licenses/by/4.0/ (accessed on 10 January 2025).

**Figure 3 nutrients-17-01319-f003:**
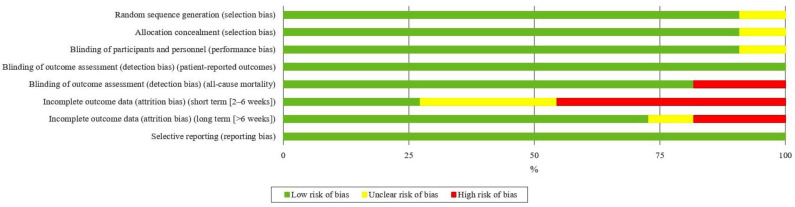
Graph of risk of bias. Green represents a negligible risk of bias, yellow represents an unclear risk, and red represents a substantial risk of bias. The overall risk of bias for each category is shown. For example, the size of the green rectangle corresponds to the number of studies deemed to have a minimal risk of bias.

**Figure 4 nutrients-17-01319-f004:**
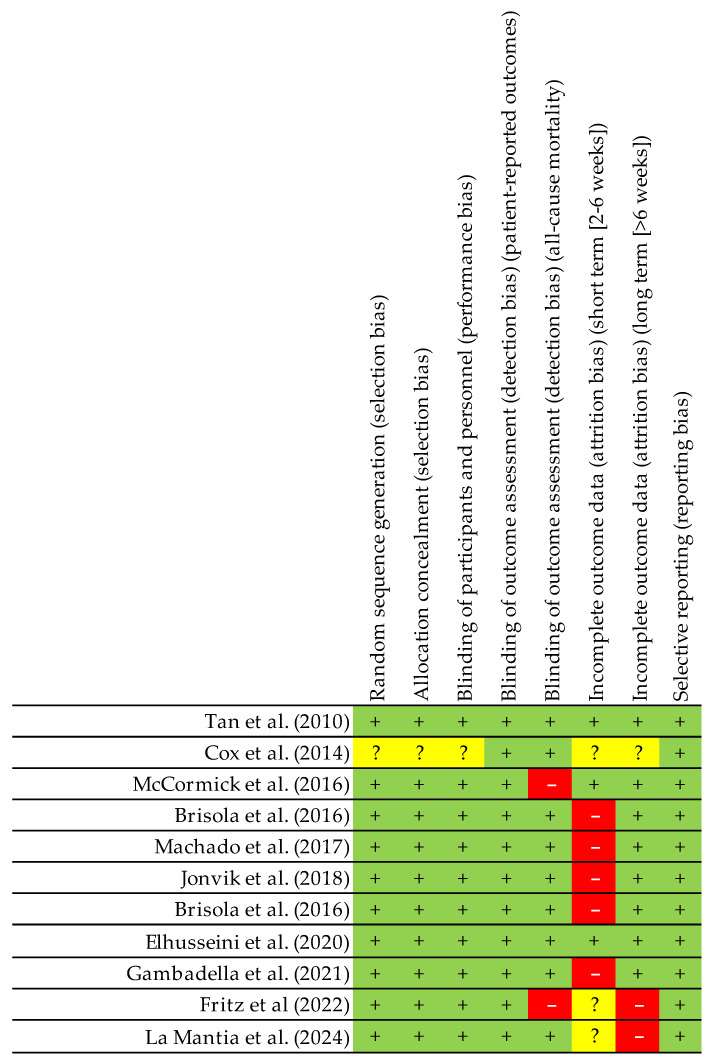
Summary of risk of bias, indicating the risk of bias for each domain in each study. Red: high level of bias; green: low level of bias; yellow: unclear level of bias; +: possibility of low bias; -: possibility of high bias; ?: undetermined bias [43,44,45,46,47,48,49,50,51,52,53].

**Table 1 nutrients-17-01319-t001:** Levels of evidence of the selected studies according to the Oxford quality scoring system [46].

Study	Level of Evidence
Tan et el. (2010) [53]	1B
Cox et al. (2014) [25]	1A
McCormick et al. (2016) [54]	1B
Brisola et al. (2016) [55]	1B
Claus et al. (2017) [56]	1B
Jonvik et al. (2018) [57]	1B
Brisola et al. (2018) [58]	1B
Elhusseini et al. (2020) [59]	1B
Gambardella et al. (2021) [60]	1B
Fritz et al. (2022) [61]	1B
La Mantia et al. (2024) [62]	1B

1A: systematic review (with homogeneity) of level 1 diagnostic studies or clinical decision rule with 1B studies from different clinical centres; 1B: double-blind comparison of an appropriate spectrum of consecutive patients, all of whom have undergone the diagnostic test and the reference standard.

**Table 2 nutrients-17-01319-t002:** Studies included in the review.

Scope of Knowledge	Journal	Q	Authors	Population	Age Ranges (Years)	Method	Intervention	Variables	Outcomes Analysed	Main Conclusions
SM	Int J Sport Nutr Exerc Metab	1	Tan et el. (2010) [53]	12 ♀ EWPp	23.7 ± 3.0	DB	1 week	NaHCO_3_	Antrop; MTS; 4 × 10 MSs; BL; PER; Bf; Ad; HR	Average sprint performance
SM	Int J Sport Nutr Exerc Metab	1	Cox et al. (2014) [25]	♂ and ♀ EWPp	*-*	DB	-	PD; DI; BC; N; SSFs	TMA; PDDC; FDC; DI; BC; NS; H	Uses all metabolic pathways in ♂ HIs is + important for ♀ in the game area In central ♀ players are + important in the fight _max_FC above 80% in ♂ aFCr in ♀ is 80% Gd high due to sustained intensity Ss deteriorates with fatigue Di Pt higher in ♂ tan in ♀ ♂ balanced mesomorphs ♀ endomorphs. HMB does not influence S and FFM use of NaHCO_3_ is beneficial in en ♀ CM no clear benefit in ♂
FS	J Int Soc Sports Nutr	2	McCormick et al. (2016) [54]	9 ♂ EWPp	18.6 ± 1.4	DB	7 days	sCJ with [high] of PhCs and Atc	IL-6; CRP; Ua; F2-IsoP; TQR; DOMS	Recovery of performance
M	PLoS ONE	1	Brisola et al. (2016) [55]	22 ♂ EWPp	18 ± 4	DB	4 weeks	RSA; PER; BL; 30 m s	β-a	Slight improvement for RSA
SM	Pediatr Exerc Sci	1	Claus et al. (2017) [56]	15 ♂ EWPp	16 ± 2	DB	6 weeks	Wt; Mdtx; β-a	RSAwft; Mx; 200-m s; BL	Performance
SM	Int J Sport Nutr Exerc Metab	1	Jonvik et al. (2018) [57]	14 ♀ EWPp	22 ± 4	DB	4 weeks	DAt; Ist	Nitrate	Performance in Ist
										DAt
Me	PLoS ONE	1	Brisola et al. (2018) [58]	22 ♂ EWPp	18 ± 4	DB	4 weeks	β-a	VO_2pmax_; SVO_2pmax;_ 3AO	SVO_2pmax_
										VO_2pmax_
SM	BMJ Open Sport Exerc Med	2	Elhusseini et al. (2020) [59]	12 ♂ EWPp	18–22	DB	2 measurements on a CE	P G	BC; HR; ee; EE; RQ; RR; pt	P does not affect ee P does not affect EE Significant increase in the HR. No significant differences in the eE.
Me	Oxid Med Cell Longev	1	Gambardella et al. (2021) [60]	17 ♂ EWPp	29.3 ± 1.66	DB	4 weeks	MsS; BL; IGF1; CPK; MPul:	L-Arginine	Performance
SM	BMC Sports Sci Med Rehabil	3	Fritz et al. (2022) [61]	19 ♂ EWPp	18–34	DB	4 meses	BC Antrop Nh	BMC; EW; SMM; BcM IW; FFM; TBW; BMR; BfM; WBPA; RBC; Hm; RBCn; G; Cr; GGT; GPT; GOT; F; Lk; U; CK; Ua; Hg; Ch; HDL; LDL; Tg; Na; K; Mg; Ca; vD	Di Pt and CHO are much lower than the recommendations opt DI Pt is before and after training, and regularly 3 to 5 times a day High [K] for big tV U ll vD ll associated with overload
FS	Nutrients	1	La Mantia et al. (2024) [62]	20 ♂ EWPp	18–35	DB	8 weeks	Spirulina + Cu	ASPS CPK	ASPS Performance Muscle tension

Legend: []: concentration; ♀: female; ♂: male; 3AO: 3 min all-out; Ad: abdominal discomfort; aFC_r_: average heart rate reserve; Antrop: anthropometry; ASPS: Athlete’s Subjective Performance Scale; Atc: anthocyanin; BC: body composition; BcM: body cell mass; Bf: bowel fullness; BfM: body fat mass; BL: blood lactate; BMC: bone mineral content; BMR: basal metabolic rate; Ca: calcium; CE: cycle ergometer; Ch: cholesterol; CHO: carbohydrate; CK: creatine kinase; CM: creatine monohydrate; CPK: creatine phosphokinase; Cr: creatinine; CRP: C-reactive protein; Cu: copper; DAt: dynamic apnoea test; DB: data-based; Di: daily intake; DI: dietary intake; DOMS: delayed-onset muscle soreness; EE: efficiency of the exercise; eE: energetic efficiency; ee: energy expenditure; EW: extracellular water; EWPp: elite WP players; F: ferritin; F2-IsoP: F2-isoprostane; FDC: fatigue during competition; FFM: fat-free mass; FS: food science; G: glucose; Gd: glycolytic demand; GGT: gamma-glutamyl transferase; GOT: glutamate oxaloacetate transaminase; GPT: glutamate pyruvate transaminase; H: hydration; HDL: high-density lipoprotein; Hg: haemoglobin; HIs: high-intensity swimming; Hm: haematocrit; HMB: b-hydroxy-b-methylbutyrate; HR: heart rate; IGF1: insulin-like growth factor; IL-6: interleukin 6; Ist: intermittent sprint test; IW: intracellular water; K: potassium; LDL: low-density lipoprotein; Lk: leucocyte; ll: low level; Me: medicine; M: minutes; _max_FC: maximum heart rate; Mdtx: maltodextrin; Mg: magnesium; MPul: maximum power of the upper limbs; MSs: maximal-sprint swims; MsS: maximum swimming speed; MTS: match-simulation test; Mx: maximal 30 s tethered swimming in alternate eggbeater kick; N: nutrition; Na: sodium; NaHCO_3_: sodium bicarbonate; Nh: nutritional habit; NS: nutritional strategy; opt: optimal time; P: phosphorus; PD: physiological demand; PDDC: physiological demand during competition; PER: perceived effort rating; PhCs: phytochemicals; pt: perceived tiredness; Pt: protein; RBC: red blood cell sink; RBCn: red blood cell number; red: detrimental effect; RQ: respiratory quotient; RR: respiratory rate; RSA: repeated-sprint ability; RSAwft: specific WP repeated-sprint ability with free throw; S: strength; s: swimming; sCJ: sour cherry juice; SM: sport medicine; SMM: skeletal muscle mass; Ss: shooting skill; SSFs: supplements and sports foods; SVO_2pmax_: strength associated with VO_2pmax_; TBW: total body water; Tg: triglyceride; TMA: time–motion analysis; TQR: total quality of recovery; tV: training volume; U: urea; Ua: uric acid; vD: vitamin D; green: favourable effect; VO_2pmax_: peak maximum oxygen uptake; WBPA: whole-body phase angle; Wt: whey protein; yellow: effect not substantial or not clear; β-a: β-alanine.

## Data Availability

This work is a review. Therefore, the information in this document is open to all readers.

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
