# Peer review of "Ergonutrition Supplementation and Recovery in Water Polo: A Systematic Review"

_nutrients, 2025, doi:10.3390/nu17081319_

Round 1

Reviewer 1 Report (Previous Reviewer 2)

Comments and Suggestions for Authors

The review proffers a poorly-justified role for beta-alanine and L-arginine as nutritional supplements for elite water polo (WP) players though no recommendations are made at this stage of research.

I remain concerned the authors cite many papers that are not specific to WP but refer to team sports or water sports in general. The supplements tested in WP are few and all studies described were of very low quality.

Can the authors please discuss the positives and negatives of applying generic nutrition advice and education to the water polo athlete and the evidence for that discussion? In the second paragraph of the introduction I counted 2 references to water polo and 17 generic references. I repeat "What are the similarities and differences (and lessons to be learnt from the nutritional literature) between WP and other anaerobic/aerobic team sports?"

Author Response

Reviewer 1

Reviewer The review proffers a poorly-justified role for beta-alanine and L-arginine as nutritional supplements for elite water polo (WP) players though no recommendations are made at this stage of research.
Authors: Thanks so much, good point. We have expanded the justification for beta-alanine and L-arginine as nutritional supplements for elite water polo players (WP) following your instructions (lines 696-745, in red in the text).

Reviewer: I remain concerned the authors cite many papers that are not specific to WP but refer to team sports or water sports in general. The supplements tested in WP are few and all studies described were of very low quality.
Authors: Agree with your proposal. We have revised the articles used, eliminating/changing those that are from other non-specific sports disciplines, similar to or related to the WP. We hope you like the changes made.

Reviewer: Can the authors please discuss the positives and negatives of applying generic nutrition advice and education to the water polo athlete and the evidence for that discussion? In the second paragraph of the introduction I counted 2 references to water polo and 17 generic references. I repeat "What are the similarities and differences (and lessons to be learnt from the nutritional literature) between WP and other anaerobic/aerobic team sports?"
Authors: following your accurate indications, we have proceeded to change references to sports not related to WP and have also added the similarities and differences between water polo and other anaerobic/aerobic team sports (lines 108-129, in red in text).

Reviewer 2 Report (New Reviewer)

Comments and Suggestions for Authors

This great systematic review examines nutrition and nutrients in water polo. The article is structured and comprehensive. I have some remarks:

  • Table 2: the number of different journals appears to be limited; does this yield a form of bias?
  • Table 3: the risk of bias is generally low; could a meta-analysis be possible anyway?
  • Discussion: sportspeople typically use much more supplements than noted in this paper, which should be discussed.

Author Response

Reviewer 2

Reviewer This great systematic review examines nutrition and nutrients in water polo. The article is structured and comprehensive. I have some remarks:
•    Table 2: the number of different journals appears to be limited; does this yield a form of bias?
Authors: Thanks so much for your words. We understand that this is not a form of bias, since the publication with the highest representation (International Journal of Sport Nutrition and Exercise Metabolism; 27.3%) is a specific journal on the subject, indexed in Q1, known for its great relevance and seriousness, which disseminates original scientific studies and academic analyses. It also includes articles on the application of biochemistry, physiology and nutrition to sport. The magazine also presents opinions, summaries of articles from other fields, research notes and reviews of books, videos and other announcements. Similarly, another of the most represented journals (9.1%) is the Journal of the International Society of Sports Nutrition, which focuses on how sports nutrition and supplements affect the body in the short and long term, including its composition, performance and energy expenditure. This journal aims to bring together researchers and people interested in sport to share information on how exercise and nutrition affect health, illness, recovery, training and sports performance. And the rest are diverse, all of high quality and indexed in the major quality indexes.
•    Table 3: the risk of bias is generally low; could a meta-analysis be possible anyway?
Authors: Thanks, you are ok. We have added a narrative meta-analysis on the effect of nitrates and beta alanine on WP (lines 507-560, in red in text).

•    Reviewer. Discussion: sportspeople typically use much more supplements than noted in this paper, which should be discussed.
Authors: Yes, sorry for that, based on your comment,we have briefly addressed this aspect following their indications (lines 685-691, in red in the text).

Reviewer 3 Report (New Reviewer)

Comments and Suggestions for Authors

-   The review includes only 11 studies (Line 299), which may limit its generalizability. Please clarify if inclusion criteria were excessively restrictive or if this reflects the scarcity of research in this field. Consider including studies from closely related aquatic sports (e.g., swimming, synchronized swimming) for context.

- Absence of Meta Analysis 

   While heterogeneity is acknowledged, the manuscript would benefit from a quantitative synthesis or narrative meta-analysis (Lines 288-294). This would strengthen the review’s impact by offering effect size estimations for supplements discussed.

-Gender Considerations 

   The underrepresentation of female water polo players in the included studies (Line 55) is notable. Expand your discussion (Lines 436-439) to address how future research could address this gap and improve gender specific recommendations.

-   The concept of periodized nutrition is briefly mentioned (Line 471). I recommend elaborating on this by discussing how nutritional demands change during season, offseason, and tapering phases, supported by recent references.

-   The risk of bias is well documented in tables (Lines 343-357), but please integrate a clearer narrative on how these limitations influence the strength of your conclusions (e.g., near Lines 544-545).

 Minor Comments

- Line 51: Include more explanation or references regarding why NaHCO3 benefits females specifically, while creatine shows no clear benefit for men.

- Line 104: Rephrase "comprehensive analysis focused on aspects related to WP" to "systematic evaluation of nutritional and ergogenic strategies in water polo recovery."

- Line 164: The PRISMA diagram should be updated to comply with the 2020 PRISMA checklist format.

- Line 297: Clarify how studies were determined to be “unrelated” (e.g., via keyword exclusions, screening procedures).

- Line 308: Correct "cherry pie" to "sour cherry juice" for accuracy.

- Line 433: "To help coaches and improve player performance" could be more formally phrased as "to assist coaches in applying evidence-based strategies for enhancing player performance."

- Line 476: Expand the section on nutritional periodization by referencing Stellingwerff et al., 2019 (already cited in references).

- Line 531: Consider using "gastrointestinal discomfort" instead of "gastrointestinal distress" for consistency.

- Line 594: Ensure author contributions comply with your journal’s guidelines, e.g., using CRediT taxonomy.

-Line 613: The references section appears truncated. Ensure all references are complete and formatted according to journal style.

Once these points are addressed, the manuscript should be suitable for publication.

Author Response

Reviewer 3

Reviewer. The review includes only 11 studies (Line 299), which may limit its generalizability. Please clarify if inclusion criteria were excessively restrictive or if this reflects the scarcity of research in this field. Consider including studies from closely related aquatic sports (e.g., swimming, synchronized swimming) for context.
Authors: We have responded to this question in more detail in the text (lines 356-378, in red in the text). In our review protocol, we have placed particular emphasis on ensuring the objectivity and transparency of the study selection process. To this end, we have defined explicit and unequivocal inclusion criteria that rigorously guide the selection of relevant literature. We consider that the inclusion criteria specified in lines 169 to 172 of our protocol are not only reasonable, but also sound and justified, being based on robust scientific arguments and the best available evidence. These criteria act as the fundamental basis on which the selection process is built, clearly and consistently dictating whether a specific study will be included in the systematic review we propose.
In addition, recognising the importance of avoiding any potential grey area or ambiguity in the evaluation of the studies, we have incorporated equally explicit and detailed exclusion criteria, which can be found defined on lines 173 to 177 of the document. The inclusion of these exclusion criteria seeks to offer greater clarity and precision to the process, eliminating any margin of subjective interpretation that could compromise the validity and reproducibility of the results.
It is crucial to emphasise that the inclusion criteria have not been developed or considered in isolation. On the contrary, they have been carefully articulated so that they are as mutually exclusive as possible, minimising the overlap and redundancy of information. This meticulous design ensures that each criterion focuses on a specific aspect of the studies, avoiding the duplication of information relevant to other elements of the PICO framework (Population, Intervention, Comparison, Outcome) that guides our research. 
We have carried out an exhaustive review using 11 studies that represent the current state of the art in this field. While we recognise that the number of papers analysed, 11 in total, might seem modest, it is important to highlight that there are published and recognised systematic reviews within the scientific literature that are based on an even smaller number of articles. As an example:
•    Mielgo-Ayuso J, Calleja-Gonzalez J, Marqués-Jiménez D, Caballero-García A, Córdova A, Fernández-Lázaro D. Effects of Creatine Supplementation on Athletic Performance in Soccer Players: A Systematic Review and Meta-Analysis. Nutrients. 2019 Mar 31;11(4):757. doi: 10.3390/nu11040757. PMID: 30935142; PMCID: PMC6520963.
•    Serafim TT, Oliveira ES, Migliorini F, Maffulli N, Okubo R. Return to sport after conservative versus surgical treatment for pubalgia in athletes: a systematic review. J Orthop Surg Res. 2022 Nov 11;17(1):484. doi: 10.1186/s13018-022-03376-y. PMID: 36369155; PMCID: PMC9652835.
•    Jones BA, Arcelus J, Bouman WP, Haycraft E. Sport and Transgender People: A Systematic Review of the Literature Relating to Sport Participation and Competitive Sport Policies. Sports Med. 2017 Apr;47(4):701-716. doi: 10.1007/s40279-016-0621-y. PMID: 27699698; PMCID: PMC5357259.
Therefore, although we appreciate your observation regarding the size of the sample, we consider that the rigorous methodology employed and the relevance of the selected studies justify the validity and value of our review. Furthermore, we would like to emphasise that the quality of the studies analysed, in terms of design, methodology and results, has been a primary criterion in the selection, prioritising those works that offer an in-depth and up-to-date view of the subject in question.
This systematic review, based on the exhaustive selection of eleven studies that rigorously met the previously defined inclusion and exclusion criteria for the sport discipline in question, highlights a significant limitation in the availability of data and research specifically focused on this area. This paucity of scientific literature suggests an area of study that is relatively unexplored or with areas that still require greater research attention.
Due to the limited number of studies analysed, the conclusions drawn from this review should be interpreted with caution and cannot be generalised to the population as a whole or to broader contexts. The external validity of the findings is compromised by the limited sample, which means that the conclusions must be considered preliminary and subject to confirmation by future research with larger and more diverse samples. 
This limitation is addressed in the limitations section (lines 808-815, in red in text).

- Reviewer Absence of Meta Analysis 
   While heterogeneity is acknowledged, the manuscript would benefit from a quantitative synthesis or narrative meta-analysis (Lines 288-294). This would strengthen the review’s impact by offering effect size estimations for supplements discussed.
Authors: we have added a narrative meta-analysis on the effect of nitrates and beta alanine on WP (lines 507-560, in red in text).

- Reviewer Gender Considerations 
   The underrepresentation of female water polo players in the included studies (Line 55) is notable. Expand your discussion (Lines 436-439) to address how future research could address this gap and improve gender specific recommendations.
Authors: Yes, We have expanded the discussion (lines 571-605, in red in the text) following your instructions.

-   Reviewer The concept of periodized nutrition is briefly mentioned (Line 471). I recommend elaborating on this by discussing how nutritional demands change during season, offseason, and tapering phases, supported by recent references.
Authors: Ok, under your suggestions, We have explored the question of perioral nutrition in greater depth following your instructions (lines 645-661, in red in the text).

-   The risk of bias is well documented in tables (Lines 343-357), but please integrate a clearer narrative on how these limitations influence the strength of your conclusions (e.g., near Lines 544-545).
Authors: a (hopefully) clearer narrative has been included on how limitations on bias influence the strength of the conclusions (lines 796-807, in red in the text).

Minor Comments

- Line 51: Include more explanation or references regarding why NaHCO3 benefits females specifically, while creatine shows no clear benefit for men.
Authors: we have added more explanations or references as to why NaHCO3 specifically benefits women, while creatine does not show any clear benefit for men following its instructions (lines 52-86, in red in the text).
- Line 104: Rephrase "comprehensive analysis focused on aspects related to WP" to "systematic evaluation of nutritional and ergogenic strategies in water polo recovery."
Authors: line 162 has been reworded following their instructions (in red in the text).
- Line 164: The PRISMA diagram should be updated to comply with the 2020 PRISMA checklist format.
Authors: we have updated the PRISMA 2020 checklist (lines 220-222, in red in the text), following your instructions.
- Line 297: Clarify how studies were determined to be “unrelated” (e.g., via keyword exclusions, screening procedures).
Authors: we have explained how it was determined that the studies were not related as you have suggested (lines 356-378 in red in the text)
- Line 308: Correct "cherry pie" to "sour cherry juice" for accuracy.
Authors: we have changed all the indications of cherry pie to sour cherry juice as you indicate (lines 485 and 503, in red in the text).
 - Line 433: "To help coaches and improve player performance" could be more formally phrased as "to assist coaches in applying evidence-based strategies for enhancing player performance."
Authors: we have changed the wording as you indicate (line 564-565, in red in the text).
- Line 476: Expand the section on nutritional periodization by referencing Stellingwerff et al., 2019 (already cited in references).
Authors: we have explored the question of perioral nutrition in greater depth following your instructions (lines 645-661, in red in the text).
- Line 531: Consider using "gastrointestinal discomfort" instead of "gastrointestinal distress" for consistency.
Authors: we have changed discomfort to inconvenience as you indicate (line 758, in red in the text).
- Line 594: Ensure author contributions comply with your journal’s guidelines, e.g., using CRediT taxonomy.
Authors: we have completed the functions of some of the authors (lines 841 and 843, in red in the text), but we have always followed the instructions and the publication template. 
-Line 613: The references section appears truncated. Ensure all references are complete and formatted according to journal style.
Authors: we have checked the bibliography and it appears to be correct without any breaks with all the references in the text, we don't know what could have happened.

Once these points are addressed, the manuscript should be suitable for publication.

Reviewer 4 Report (New Reviewer)

Comments and Suggestions for Authors

In the manuscript submitted to me for review entitled "ErgoNutrition supplementation and recovery in water polo. A systematic review“ the authors Álvaro Miguel-Ortega, Josu Barrenetxea-García, María-Azucena Rodríguez-Rodrigo, Enrique García-Ordóñez, Juan Francisco Mielgo-Ayuso and Julio Calleja-González present an in-depth study summarizing the available information on the importance of ergonutrition and the intake of nutritional supplements in the recovery of water polo (WP) players.

My remarks and recommendations to the authors are:

  1. References from number 40 onwards in the text are presented in superscript. This is not necessary. Let's go back to the original view.
  2. Under table 1, it should be indicated what "Level of evidence 1B and 1A" means.
  3. In table 2, in the first column, it is not clear what the abbreviations (S, M, FS, Me) mean - it should be presented at the top as the column title.
  4. In the References section, reference number 8 does not have a year of publication. Let's add it.

Author Response

In the manuscript submitted to me for review entitled "ErgoNutrition supplementation and recovery in water polo. A systematic review“  the authors Álvaro Miguel-Ortega, Josu Barrenetxea-García, María-Azucena Rodríguez-Rodrigo, Enrique García-Ordóñez, Juan Francisco Mielgo-Ayuso and Julio Calleja-González present an in-depth study summarizing the available information on the importance of ergonutrition and the intake of nutritional supplements in the recovery of water polo (WP) players.

My remarks and recommendations to the authors are:

1.    Reviewer. References from number 40 onwards in the text are presented in superscript. This is not necessary. Let's go back to the original view.
Authors: we have removed the superscript from the references in the body of the paper.

2.    Reviewer Under table 1, it should be indicated what "Level of evidence 1B and 1A" means.
Authors: ok, thanks for this detail. The meaning of the levels of evidence has been added (lines 339-350, in red in the text).

3.    Reviewer In table 2, in the first column, it is not clear what the abbreviations (S, M, FS, Me) mean - it should be presented at the top as the column title.
Authors: Good idea. The title of the column has been included, although these abbreviations are explained in the table footnote.

4.    Reviewer In the References section, reference number 8 does not have a year of publication. Let's add it.
Authors: OK, this reference has been updated (line 897, in red in the text).

Round 2

Reviewer 1 Report (Previous Reviewer 2)

Comments and Suggestions for Authors

.

This manuscript is a resubmission of an earlier submission. The following is a list of the peer review reports and author responses from that submission.

Round 1

Reviewer 1 Report

Comments and Suggestions for Authors

Congratulations to the authors for their work. One suggestion is that the introduction should specify the percentage of technical and physical work involved in water polo

In the discussion and results sections, I suggest that the authors emphasize the supplements with the strongest scientific evidence, as per the IOC document

Comments on the Quality of English Language

Adequate

Reviewer 2 Report

Comments and Suggestions for Authors

This paper addresses the area of nutritional support of the water polo (WP) player. WP is a team sport that requires players to use a combination of aerobic and anaerobic metabolism. The paper originally had three aims (line 101): organise nutritional strategies to optimise performance of WP athletes, evaluate ergogenic aids in WP performance and identify which strategies are effective in periods of recovery. The latter aim had disappeared by the end of the paper (line 332).

The authors conclude that nutritional education is lacking in WP players and more attention must be paid to recovery. These two points are not well supported by the stated scope of the paper although the authors reference their assertions.

The reader does not emerge with a good grasp of the quality of the literature. The dietary intervention studies reported are few and small (most < 20) and the authors move between citing literature that is based upon all team sports that mingle aerobic and anaerobic exercise and literature that refers to WP players only.

Minor points:

Meaning of 5:2 on line 48

It is unnecessary to state “to the authors’ knowledge” (line 59). Position the apostrophe too.

Lines 60 to 72: This paragraph muddles all aerobic/anaerobic team sports with WP. What are the similarities and differences (and lessons to be learned from the nutritional literature) between WP and other aerobic/anaerobic team sports?

Line 97-99 repetitive

I am insulted by the reference to toilet paper on line 216/7. I do not appreciate my time being wasted.

Figure 3 the numbers do not add up (eg 11+112 = 123)

This is a challenging series of papers to condense into a review. Narrative would be better than systematic. Small studies on disparate supplements with different outcomes (some of those outcomes are more subjective than objective). The table is not easy to read even with the copious footnote of abbreviations explanations

Age ranges might help

Line 321 “As did” conveys ambiguity. Make it clear there were no performance enhancing effects of these supplements.

Section 3.4 is missing.

Line 332. What about the third objective?

Line 338. Need references for these statements

Lines 348 and 350 are repeated. It would help if I were not cast in the role of a proof reader.

Line 363 “These articles” but only one is mentioned

Lines 455 and onwards are poor English. Possibly once bulleted but poorly converted to text.

The conclusions are not well-aligned with the objectives and the criticism or interpretation of the tabled papers is unhelpful and superficial. (Small studies on a variety of agents). I am still not sure what differences exist between a WP player’s recovery approach and nutritional needs and those of a player in another discipline.

Comments on the Quality of English Language

See comments on English usage above.

Reviewer 3 Report

Comments and Suggestions for Authors

The manuscript entitled “ErgoNutrition supplementation and recovery in water polo. A systematic review.” presents information of some type of supplementation in the recovery of Water polo players.

The first sentence in the results section of the abstract is not directly related to the topic of this review.
Table 1 needs to be reorganized so that the column titles are smaller. Additionally, it is necessary to explain the meaning of the symbols (below the table), for example, what the question mark signifies.
‘Assessing the Quality of Experiments: Risk of Bias and Levels of Evidence’ – this subsection should contain only the names and a detailed description of the criteria for evaluating the articles. The actual results of this evaluation should be presented in the results section of the article.
“Oxford quality scoring system” - This must be written precisely in the methodology (more information is needed)—what the evaluation criteria are and what the individual values mean. However, the results of this analysis (table 2) should be in the Results section, not in the methodology.
It is not clear why three different criteria were used to evaluate the articles. What does this mean for the review? This should be explained. Please consider whether all three are truly necessary.
The flow chart should be in the methodology, not in the results.
What does "Related to other disciplines" mean on the flowchart? This is not present in the original document—why is it there? What does it signify?
On the flowchart, "Record" appears once and "Reports" at another point—these are two different things. Please check the original document. Please correct this.
Table 3 needs to be reformatted. What do they signify? The captions under the table are too long. The table needs to be simplified. Why are there colors? The meaning of the colors is explained below the table, but it might be helpful to add a column and describe the effect in that column. This would make it easier to read.
In this version, Table 2 is completely unreadable, and it is difficult to extract the most important information from it. Perhaps it should be divided into two tables in a horizontal orientation, and fewer abbreviations should be used to make the text more understandable for the reader.
‘3.4. Figures, Tables and Scheme’ - Something is missing here.
There is a lack of a simple table summarizing the articles included in this review.
The conclusions are too broad, especially since this systematic review focuses only on supplementation and not on overall nutrition.
The title of the article should reflect the content, so there are no associations with a specific company but rather with a group of products, especially since the articles included in the analysis are diverse in this regard.